# Inkjet-Printed Dielectric Layer for the Enhancement of Electrowetting Display Devices

**DOI:** 10.3390/nano14040347

**Published:** 2024-02-12

**Authors:** Hongwei Jiang, Rongzhen Qian, Tinghong Yang, Yuanyuan Guo, Dong Yuan, Biao Tang, Rui Zhou, Hui Li, Guofu Zhou

**Affiliations:** 1Guangdong Provincial Key Laboratory of Optical Information Materials and Technology, Institute of Electronic Paper Displays, South China Academy of Advanced Optoelectronics, South China Normal University, Guangzhou 510006, China; 2021010258@m.scnu.edu.cn (H.J.); 2021024056@m.scnu.edu.cn (R.Q.); tinghongyang@m.scnu.edu.cn (T.Y.); guoyy@scnu.edu.cn (Y.G.); yuandong@scnu.edu.cn (D.Y.); tangbiao@scnu.edu.cn (B.T.); guofu.zhou@m.scnu.edu.cn (G.Z.); 2College of Mechatronics and Control Engineering, Shenzhen University, Nanhai Ave. 3688, Shenzhen 518060, China

**Keywords:** electrowetting display (EWD), inkjet printing, dielectric layer, photoresist, leakage current

## Abstract

Electrowetting with a dielectric layer is commonly preferred in practical applications. However, its potential is often limited by factors like the properties of the dielectric layer and its breakdown, along with the complexity of the deposition method. Fortunately, advancements in 3D inkjet printing offer a more adaptable solution for making patterned functional layers. In this study, we used a negative photoresist (HN-1901) to create a new dielectric layer for an electrowetting display on a 3-inch ITO glass using a Dimatix DMP-2580 inkjet printer. The resulting devices performed better due to their enhanced resistance to dielectric breakdown. We meticulously investigated the physical properties of the photoresist material and printer settings to achieve optimal printing. We also controlled the uniformity of the dielectric layer by adjusting ink drop spacing. Compared to traditional electrowetting display devices, those with inkjet-printed dielectric layers showed significantly fewer defects like bubbles and electrode corrosion. They maintained an outstanding response time and breakdown resistance, operating at an open voltage of 20 V. Remarkably, these devices achieved faster response times of ton 22.3 ms and toff 14.2 ms, surpassing the performance of the standard device.

## 1. Introduction

The elucidation of electrowetting behavior on surfaces with varying charges, initially observed by Gabriel Lippmann in 1875 [1]—though noted earlier in observations—marked a significant milestone. A. N. Frumkin’s 1936 demonstration further expanded this understanding by showcasing the capacity to manipulate water drop shapes through surface charge [2]. Lippmann’s work in 1875 underscored the electrocapillary action’s potential to alter liquid droplet shapes via voltage application. Subsequent research has delved into comprehending and refining droplet actuation and contact angle behavior. Present investigations of electrowetting on dielectric (EWOD) focus on the reduction of power consumption, the exploration of diverse dielectric materials, and the optimization of device structures.

Since Lippmann’s pioneering work, extensive scrutiny and the application of electrowetting have spanned multiple domains, encompassing electronic paper displays [3,4,5], variable focus lenses [6,7,8], and lab-on-chip technologies [9,10,11,12]. Electrowetting displays, particularly favored for their role as reflective displays, have garnered substantial attention. In comparison to conventional displays, electrowetting displays present numerous advantages, including diminished power consumption, vibrant full-color output, rapid video rendering, paper-like attributes, and the potential for flexible display configurations.

Typically, a standard electrowetting display comprises five necessary components: top and bottom electrodes, immiscible polar liquid and colored oil, pixel wall grid, and the fluoropolymer layer. In this setup, electrowetting phenomena occur due to the wetting and dewetting of the polar liquid on the fluoropolymer surface, regulated by an external electrical field. The performance of electrowetting display devices is significantly influenced by the characteristics of both the polar liquid and the fluoropolymer layer. While a single fluoropolymer film can function as both the dielectric and hydrophobic layer, it is not perfect for use in EWOD due to its porous nature and low breakdown resistance. Typically, existing EWOD microfluidic devices are coated with Teflon as a hydrophobic layer [13,14,15]. However, Teflon is costly and has a low breakdown voltage. Consequently, the deposition of an additional dielectric material capable of withstanding high voltage is desirable.

In current practice, an increasing trend involves the utilization of inorganic or polymeric materials as dielectric layers [16,17]. Inorganic materials, including SiO_2_ [18], Si_3_N_4_ [19], silicon nanospheres [20], and mesoporous silica [21], exhibit favorable attributes like high dielectric constants and low driving voltages. Notably, metal oxides such as ZnO [22,23,24] exhibit high polarity, leading to a reduction in dielectric efficiency. In addition, polymeric materials like Teflon AF [25,26], Cytop TM [27], Parylene-C [28], polydimethylsiloxane [29], polyimide [30], poly(ethylene terephthalate)/polyethylene [31], and photoresist Su-8 [32] can also be used as the dielectric layer. However, regardless of whether it is an organic or inorganic material, deposition methods for these materials typically involve vapor deposition, evaporation, and spin coating, all of which demand stringent process requirements and form continuous whole films. For patterned film preparation, additional etching processes are necessary, resulting in complex procedures and low material efficiency. In this paper, we utilized inkjet printing to directly prepare patterned dielectric layers using negative photoresist material in a single step. The process is straightforward and more material efficient. Although negative photoresist lacks the high dielectric constant found in inorganic dielectric materials, its uniform and pin-hole-free characteristics contribute significantly to enhancing reliability within a tolerable range of increased driving voltage.

## 2. Theory of Electrowetting and Inkjet Printing

### 2.1. Fundamentals of Electrowetting on Dielectric Layer

Following classical electrowetting theory, an external voltage between the conductive fluid and the bottom electrode leads to a reduction in the contact angle on a hydrophobic surface [33]. Specifically, when considering a sessile aqueous droplet situated upon a hydrophobic insulating surface, this phenomenon is recognized as electrowetting on dielectric, as exemplified in Figure 1. In the initial state, the liquid droplet exhibits a non-wetting condition on the surface of the hydrophobic layer, with a contact angle exceeding 90 degrees, as depicted by the solid line in Figure 1. Upon the application of a voltage between the liquid droplet and the hydrophobic layer, a modification in the wetting behavior of the liquid droplet on the hydrophobic surface occurs. This results in a reduction in the contact angle, thereby facilitating the wetting of the solid surface, as indicated by the dashed line in Figure 1. The voltage, denoted as V, exclusively influences the tension at the solid–liquid interface rather than the tensions at the solid–gas and liquid–gas interfaces. The reduction in solid–liquid interfacial tension equates to an amount equivalent to the electrostatic energy CV^2^/2, where C (F) signifies capacitance. Notably, considering the electrode’s coverage by a hydrophobic insulating layer characterized by a thickness denoted as d (m) and a specific dielectric constant, this behavior can be elucidated through the following expression:(1)$$c\mathrm{os}\theta_{V}=\frac{\gamma_{sg}-\gamma_{sl}}{\gamma_{lg}}+\frac{1}{2}\frac{{CV}^{2}}{\gamma_{lg}}=\cos\theta_{0}+\frac{{\varepsilon_{r}\varepsilon }_{0}}{2\gamma_{lg}d}V^{2}$$
where $\theta_{V}$ and $\theta_{0}$ represent the contact angles of the liquid droplet under the influence of an applied voltage V and in the absence of voltage 0, respectively. $\gamma_{sg}$ (N/m), $\gamma_{sl}$ (N/m) and $\gamma_{lg}$ (N/m) represent surface tensions of solid–gas, liquid–gas, and solid–liquid interfaces, respectively. $\varepsilon_{0}$ (F/m) and $\varepsilon_{r}$ (F/m) represent the permittivity of the vacuum and the effective permittivity of the dielectric layer, respectively.

An effective strategy to achieve enhanced electrowetting performance involves significantly increasing the dielectric’s thickness, thereby reducing its capacitance. This augmentation often entails the utilization of common polymers coated with a fluoropolymer or the adoption of a stand-alone fluoropolymer film, typically several micrometers thick. As the dielectric thickness amplifies, there is a corresponding decrease in the required electric field E (N/C) across the dielectric of thickness d, as indicated by Equation (1). Equation (2) is Equation (1) solved for voltage:(2)$$V_{EW}=\sqrt{\frac{2\gamma_{lg}\times d(cos\theta_{V}-cos\theta_{0})}{{\varepsilon_{r}\varepsilon }_{0}}}$$

As is demonstrated in Equation (3), it becomes evident from Equation (2) that the voltage necessary to induce a specific contact angle alteration is directly proportional to the square root of the dielectric thickness. It is understood that the requisite electric field for electrowetting ($E_{EW}$) is consistently determined by the potential divided by the dielectric layer thickness d.
(3)$$V_{EW}\;\propto \;\sqrt{d},\;E_{EW}=V_{EW}/d\;∴\;E_{EW}\;\propto \;\sqrt{1/d}$$

From Equation (3), it can be inferred that reducing the thickness of the hydrophobic layer can decrease the potential, theoretically enhancing the reliability of the device. In reality, achieving the same reduction in contact angle requires an increased electric field, and the thinner dielectric itself is more susceptible to catastrophic defects due to the porous nature of hydrophobic materials. Increased dielectric layer thickness commonly results in a reduction of critical defects compared to thinner counterparts. Submicroscopic flaws that could potentially affect an extremely thin dielectric might not propagate throughout the entirety of a thicker dielectric layer, thus diminishing the risk of complete dielectric failure. Therefore, within a certain range of high potentials that are not limiting factors, and with consistent energy consumption, the use of an additional dielectric layer combined with fluoropolymer proves to be a feasible approach to achieve a defect-free and reliable performance. Negative photoresist, as a photosensitive curing material, is commonly employed in the fabrication of micro and nanostructures in display panels due to its uniform film formation and diverse preparation methods. This also offers a choice for dielectric materials in electrowetting display fabrication.

### 2.2. Principle of Inkject Printable Materials

The conventional preparation method for a photoresist typically involves coating, exposure, and development to achieve the desired patterned layer, making the process relatively complex. In contrast, inkjet printing, as an additive manufacturing technique, enables the direct printing of the material into the required patterns. It has garnered widespread attention, particularly in printing display fabrications like OLED.

The fluidic parameters of an ink, such as viscosity, density, and surface tension, significantly influence the jetted droplet’s shape, size, substrate wettability, and the potential occurrence of satellite droplets, and thus determine the ink’s “jettability”. Ideally, the ink’s viscosity should be sufficiently low to ensure the expulsion of droplets from the nozzle via a transient pressure pulse and the refilling of the ink reservoir within 100 ms [34]. Moreover, the ink’s surface tension plays a crucial role: it must be high enough to prevent unwanted dripping while remaining low enough for the detachment of the ejected droplet from the nozzle [35]. Though specific values of these properties may vary across printers, in the realm of inkjet printing, recommended viscosity typically falls within the range of 1–25 mPa·s, while the surface tension ranges between 25 to 50 mN/m. These parameters serve as essential benchmarks for optimal ink performance in the printing process [36].

The impact of fluidic properties alongside variables such as nozzle diameter on the material’s “jettability” can be efficiently assessed by employing dimensionless numbers, including the Reynolds (Re), Weber (We), and Ohnesorge (Oh) numbers [37]. The Reynolds number, delineating the ratio between inertial and viscous forces, is defined as:(4)$$Re=\frac{\rho \nu L}{\eta }$$

The Weber number, representing the relationship between inertial forces and surface tension forces, is determined by the following calculation:(5)$$We=\frac{\rho \nu^{2}L}{\sigma }$$

The Ohnesorge number characterizes droplet formation, illustrating the ratio between viscous forces and the combined influence of surface tension and inertial forces, given by the equation:(6)$$Oh=\frac{\sqrt{We}}{Re}=\frac{\eta }{\sqrt{\rho L\sigma }}$$
where ρ (Kg/m^3^), η (pa·s), σ (N/m), L (m), and ν (m/s) represent the density, viscosity, surface tension, nozzle diameter, and droplet velocity, respectively. Fromm J. E. was among the pioneers in employing dimensionless numbers to assess the appropriateness of inks for droplet formation [38]. He suggested that, if the inverse of the Ohnesorge number, denoted as Z, exceeds two, stable droplet formation could be achieved. This concept was further enhanced by Reis [39], who, through numerical simulations, demonstrated that a range of values within 1 to 10 for Z ensures stable droplet generation. This also provides guidance for the selection of printable photoresists. The “jettability” of the material can be preliminarily assessed by calculating its Z-value, thus avoiding ineffective testing processes.

## 3. Device Fabrication

### 3.1. Materials

Essentially, a traditional Electrowetting Display device is constructed from two glass electrodes, a hydrophobic layer, water, and immiscible colored oil, as shown in the exploded view in Figure 2. Among them, the hydrophobic layer simultaneously functions as the dielectric layer. To distinguish it from subsequent inkjet-printed dielectric layers, we refer to it as the hydrophobic layer. In the device, the different contact angles of water and colored oil on the hydrophobic layer’s surface manifest as the non-wetting and wetting states, respectively. Therefore, in the absence of an applied voltage, colored ink spreads on the hydrophobic layer’s surface, as depicted in Figure 2a, displaying the color of the oil. Upon applying a voltage between the top and bottom electrodes, the contact angle of water on the hydrophobic layer surface decreases, transitioning from a non-wetting to a wetting state. This compels the oil to be squeezed into a corner of the pixel wall, as illustrated in Figure 2b, revealing the color of the substrate. Hydrophobic materials are typically sourced from DuPont’s AF 1600 series in the United States and Italy’s Solvay Hyflon AD series, as these fluoropolymer materials exhibit substantial water contact angles. However, due to the porous and loose characteristics of fluoropolymers, electric wetting devices experience breakdown failures after prolonged reliability testing. Therefore, we enhanced the reliability of the device by introducing an additional dense dielectric layer between the hydrophobic layer and the electrode. The negative photoresist HN-1901 has been chosen as the dielectric layer material. It, along with the negative photoresist HN-008N used in the pixel wall, was procured from Suntific Materials Co., Ltd. (Weifang, China). Additionally, Commercial 0.7 mm thick soda lime glass coated with Indium tin oxide (ITO) was purchased from Wuhu Token Technology Co., Ltd. (Wuhu, China). Colored oil was composed of Decane and dye, which was developed and synthesized by the South China Advanced Optoelectronics Institute team; its molecular structure is 1-Hydroxy-2-(2-ethylhexyl)-4-(2-ethylhexylamino)anthraquinone. We used 99% pure Decane as the solvent for the colored oil, which was sourced from Macklin Biochemical Technology Co., Ltd. (Shanghai, China). All reagents employed in this study adhere to analytical grade standards.

### 3.2. Inkjet Printing for Dielectric Layer Fabrication

The electrowetting substrate preparation was as follows: Initially, ultrasonic cleaning was conducted using 5% and 3% basic detergents, succeeded by spray cleaning with ultrapure water to eliminate any residual detergent. Following this, the substrate underwent two rounds of ultrasonic cleaning in pure water, immersion in hot ultrapure water, and final drying using hot air. It was imperative to achieve a contact angle below 25° on the Indium Tin Oxide (ITO) surface post-cleaning. Printing operations were executed utilizing a laboratory-scale inkjet printer (DMP-2850, FUJIFILM Dimatix, Inc. Santa Clara, CA, USA) with a printable range of 210 mm × 315 mm and a repeatability of ±25 μm. The printer functioned within an operational temperature range of 15–40 °C under non-condensing conditions at 5–80% relative humidity (RH). A specialized 10 pL cartridge featuring 16 nozzles, each sized at 21 μm, was chosen for dielectric material printing. Prior to the ink filling, the nozzles underwent cleaning with ethanol. The printing material needed to conform to specific criteria due to the nozzle’s limited capacity. This included a viscosity range of 10–12 mPa·s and surface tension within 28–42 mN/m at the printing temperature. Consequently, the HN-1901 material was diluted with PGMEA to 19% of its original concentration, meeting these specified requirements. The diluted HN-1901 solution was further filtered through a 1 μm filter before being introduced into the cartridge. Achieving optimal droplet shapes and material printability was facilitated by adjusting the driving voltage and waveform during the inkjet printing process. Software-based editing defined the print area, allowing the nozzles to traverse the substrate, depositing HN-1901 droplets onto the glass surface. After a brief period of surface leveling to ensure smoothness, the glass underwent pre-baking on a hot plate at 110 °C for 90 s. Subsequently, a post-baking step was essential to remove any remaining solvent residues, which was conducted at 185 °C for 30 min in an oven.

All processes were conducted within a Class 10,000 cleanroom environment to mitigate particle contamination, maintaining temperature and relative humidity at 23 ± 2 °C and 55 ± 5%, respectively.

### 3.3. Film Characterization

Various methods were utilized to characterize the inkjet-printed dielectric layer. The morphology of the printed film was examined via a microscope (AmScope 50X-2500X, United Scope, LLC., Irvine, CA, USA) and scanning electron microscopy (ZEISS Gemini 500, Carl Zeiss AG, Oberkochen, Germany). The thickness of the film was precisely measured using a profilometer (Dektak-XT, Bruker Corportaion, Billerica, MA, USA), utilizing a 2 μm probe. The probe traversed a distance of 2000 μm, applying a stylus force of 3 mg for accurate measurement.

### 3.4. EWD Device Fabrication

After the printing of the HN-1901 dielectric layer, a sequence of procedures was conducted to assemble various structures atop it, the process flow is illustrated in Figure 3a. The content within the yellow box represents the added inkjet printing process for the dielectric layer in this study, while the remaining steps constitute the conventional device fabrication process. An 800 nm fluoropolymer layer was applied over the HN-1901 layer by a spin coating process (KW-4A, Institute of Microelectronics of The Chinese Academy of Sciences, Beijing, China). Before the coating process, a 10 min UV-ozone treatment was administered to improve adhesion between the HN-1901 and the fluoropolymer material. For the preparation of the pixel walls, the inherent hydrophobicity of the fluororesin (water contact angle of about 120°) posed challenges to the direct coating of the photoresist. To overcome this challenge, the fluoropolymer coating underwent a surface modification using Reactive Ion Etching (RIE) equipment (ME-6A, Institute of Microelectronics of The Chinese Academy of Science, Beijing, China). After that, the photoresist was spin coated and the pixel wall structure was then constructed through lithography (URE-2000/35, Institute of Optics and Electronics, Chinese Academy of Science, Chengdu, China). Finally, the last step involved filling with colored oil and the assembly of the top and bottom plates.

## 4. Results and Discussion

### 4.1. Inkjet Printing Photoresist Material

The investigation of material printability holds significant importance, encompassing both functional aspects and practical applications. To ensure a high printing resolution, it is crucial to maintain specific physical properties of the ink. Excessive ink viscosity can lead to nozzle clogging, elevating shear force, and thus causing scattering or satellite spots. Thus, a key challenge in inkjet printing lies in the development of appropriate printing materials and formulations, necessitating the optimization of their physical properties to achieve precise control over the printing process.

The HN-1901 photoresist utilized in this work had a concentration of 38% and a viscosity of 50 mPa·s; the main components include acrylic resin-based photosensitive resin, a sensitizer, and a solvent, which was PGMEA. Due to the low concentration potentially resulting in an excessively thin dry dielectric film that would not effectively enhance breakdown resistance, the HN-1901 photoresist was only diluted to 19% with PGMEA solvent. This dilution reduced the viscosity to 3.783 mPa·s, which did not align with the DMP-2850 printer’s specifications. Therefore, the inkjet print head temperature was set to 45 °C to adjust the viscosity for optimal performance. During the experiment, specific parameters were precisely controlled: the peak printing voltage was maintained at 35 V, and the ejection frequency stood at 1.5 kHz, enabling the consistent generation of stable HN-1901 droplets. The physical parameters of the materials and the inkjet printing setup parameters are listed in Table 1. According to Equation (6), we can easily calculate that the Ohnesorge number is 0.24, and Z is 4.1, which meet the prerequisites for stable droplet formation. Utilizing the above parameters, the droplet formation process was simulated through ANSYS Fluent 19.2, aligning it with real-time observations during inkjet printing. For the jettability of the material, numerical simulation methods can be employed to investigate the shape and movement of ink droplets during the inkjet process. In this simulation, a Pressure-based solver was utilized with transient calculations. The velocity and pressure were solved using a Fractional Step solver, and momentum was addressed using the Quick solver. The minimum mesh quality for the inkjet printing model was set at 0.97, with a mesh partition count of 24,600 for the inkjet printing simulation. The simulation displayed remarkable congruence with the actual printed droplets, as illustrated in Figure 3a. It is noteworthy that the DMP-2580 printer’s ink cartridge incorporates 16 nozzles; the nozzle size is 21 μm, with a spacing of 256 μm. However, experimental observations revealed that neighboring ink droplets interfered with each other’s flight trajectory. Subsequently, only one nozzle was employed for the printing process. It took approximately 3 h to print a dielectric layer on a 3-inch electrowetting display device. To prevent nozzle clogging, frequent cleaning was imperative throughout the printing procedure. Consequently, to assess material printability and investigate coating thickness, a selective 1 × 1 cm^2^ pattern was printed for evaluation purposes, as illustrated in Figure 4b.

The uniformity and thickness of the dielectric layer play a pivotal role in determining the performance of the device, impacting critical aspects like driving voltage and grayscale. These factors are governed by both material attributes and printing parameters, encompassing material leveling, solvent evaporation rate, ink drop spacing, and the nozzle–substrate gap. By investigating the correlation between uniformity and drop spacing (illustrated in Figure 4b), the study aimed to understand how these factors influenced one another. While the predictability of uniformity and thickness based on drop spacing, drop size, and ink’s wettability on the substrate was acknowledged [40], their interaction was constrained by the gap between the nozzle and substrate. Notably, as the drop spacing increased, a discernible decrease in the uniformity of the dielectric layer was observed. Various drop spacings (20 μm, 25 μm, 30 μm, and 40 μm) were assessed in this experimentation. The results indicated that only the dielectric layer pattern with a spacing of 20 μm demonstrated completeness. In contrast, ink drop spacings of 25 μm and above failed to achieve a consistent and uniform film.

The film thickness data and SEM images of the dielectric layer sample with an ink drop spacing of 20 μm are presented in Figure 5. The film had an average thickness of 1.25 μm with an approximate tolerance of 16%. Notably, within a 100 μm perimeter around the pattern, a discernible coffee-ring effect was observed in the film thickness data. To accommodate this effect during the deposition process of the dielectric layer for the electrowetting display device, it is imperative to consider a 100 μm margin area. Figure 5b showcases SEM images at 70× magnifications, which highlight the absence of noticeable pin-hole defects in the film. Figure 5c is an AFM top view of the dielectric, with a scanning range of 5 by 5 μm, showing that the inkjet printed film surface is quite uniform.

### 4.2. Performance of EWDs

To evaluate the effectiveness of the inkjet-printed dielectric layer, we established a comparative analysis by creating a contrast electrowetting device that lacks a dielectric layer. This reference electrowetting display device solely utilized an 800 nm hydrophobic layer without a dielectric layer; the hydrophobic material used was DuPont AF1600, and the fabrication process is shown in Figure 3a, excluding the step of inkjet printing the dielectric layer. Both the EWD device with an inkjet-printed dielectric layer and the contrastive EWD device without inkjet printed layer shared similar specifications. These specifications included a 3-inch active area and a pixel pitch of 150 μm. This controlled experimental setup allows for a direct point-to-point comparison between the two devices, specifically focusing on the influence of the inkjet-printed dielectric layer on the overall performance of the electrowetting display. The contrastive device acts as a baseline for evaluating and quantifying the unique contributions and improvements brought about by the inkjet-printed dielectric layer. By analyzing various parameters such as driving voltage, grayscale performance, and overall display quality in both devices, we aim to gain valuable insights into the efficacy of the inkjet-printed dielectric layer. This comparative investigation will shed light on its potential to enhance the functionality and efficiency of electrowetting display devices.

In Figure 6, a comparison of the leakage current between the inkjet-printed dielectric layer EWD and the contrastive EWD is presented. Evidently, the leakage current in the inkjet-printed dielectric layer is significantly smaller than that of the contrastive EWD. This observation indicates that the inkjet-printed dielectric layer plays a crucial role in enhancing the breakdown resistance and overall reliability of the display. To further evaluate the reliability of the inkjet-printed dielectric layer, a series of reliability tests was conducted in an environment maintained at 50 °C and 50% humidity. The tests spanned durations of 24 h, 48 h, 72 h, and 96 h. Throughout these tests, the leakage current within both devices exhibited an increase with the extended duration of the reliability test. However, it is noteworthy that, even with the passage of time, the leakage current in the inkjet-printed dielectric layer device consistently remained much smaller than that of the contrastive EWD device.

This sustained lower leakage current in the device with inkjet-printed dielectric layer over the course of the reliability tests reaffirms the superior reliability and stability of the inkjet-printed dielectric layer. The dielectric layer’s ability to maintain a lower leakage current under prolonged testing conditions suggests its resilience and effectiveness in preserving the integrity of the electrowetting display. This outcome underscores the practical advantages of incorporating the inkjet-printed dielectric layer in enhancing the long-term performance and reliability of electrowetting display devices, even in challenging environmental conditions.

In Figure 7, the impact of reliability test duration on the appearance of visible defects in both devices is evident. Notably, the traditional device exhibited defects at a faster rate and with a higher frequency compared to the device with the inkjet-printed dielectric layer. Examining Figure 7a, two primary types of defects are identified: bubbles and electrode corrosion. These defects find correspondence in Figure 7b,c, respectively. The hydrophobic layer, characterized by a porous structure with pin-hole defects, plays a crucial role in this scenario. As the leakage current increases, water molecules can traverse the pinholes and come into contact with the electrodes. This interaction initiates the electrolysis of water, resulting in the generation of hydrogen gas, as illustrated in Figure 7b. The occurrence of bubbles can lead to the rupture of the hydrophobic layer. Consequently, the lower electrode becomes fully exposed to water, creating a short circuit and facilitating electrode corrosion, as depicted in Figure 7c. It is important to note that, while the inkjet-printed dielectric layer does not entirely eliminate breakdown resistance defects, it significantly improves the overall reliability of the device.

By mitigating the occurrence and impact of defects such as bubbles and electrode corrosion, the inkjet-printed dielectric layer contributes substantially to enhance the device’s reliability. This improvement, even in the face of visible defects, underscores the practical effectiveness of the inkjet-printed dielectric layer in addressing and minimizing reliability issues associated with breakdown resistance.

The capacitance characteristics of an electrowetting display (EWD) are crucial for understanding its performance, and its capacitance behaves like a complex series shunt capacitor. In the initial “Off” state, the capacitance is primarily determined by the thickness of the oil, resulting in a relatively small capacitance value. However, in the “On” state, the capacitance is influenced by both the hydrophobic layer and the dielectric layer, leading to a significant increase in capacitance. An impedance analyzer (WK6500B, Wayne Kerr Electronics, London, United Kingdom) was employed to assess the capacitance behavior of the EWDs. The capacitance–voltage (C-V) data are illustrated in Figure 8.

After subjecting the devices to various time reliability tests, the C-V curves were analyzed to evaluate the impact on capacitance. Notably, the C-V curves exhibited good repeatability, indicating consistent performance even after reliability testing. In Figure 8a, the “ON” voltage of the contrastive device is specified as 16 V, with the “Off” voltage at 10 V. Conversely, in Figure 8b, the “On” voltage of the inkjet-printed dielectric layer device is 20 V, which is 4 V higher than that of the traditional device. This increase is attributed to the introduced dielectric layer, which divides the voltage applied to the entire device. As a result, a higher voltage is required to induce the rupture of the oil film. The turn-off voltage remains consistent with traditional devices at 10 V.

Response time is a crucial parameter for assessing the performance of electrowetting displays as it directly influences the ability to display video content. In this experiment, response times were obtained by measuring the brightness changes in the EWDs, using a colorimeter (Admesy-arges45). The “On” time (t_on_) was defined as the time required to reach 90% of the maximum brightness, while the “Off” time (t_off_) was the time needed to drop from 90% of the maximum brightness to the minimum value. Figure 9 illustrates the brightness–time curves of the two devices, both driven at a voltage of 30 V. The brightness increases with time, reaching a maximum value after a certain duration and remaining constant thereafter. Upon removing the voltage, the brightness gradually decreases with time until the devices are completely closed. Figure 9a illustrates the response time of a contrastive EWD, where t_on_ is measured at 51.2 ms, and t_off_ is measured at 17.6 ms. In contrast, Figure 9b presents the response time of inkjet-printed dielectric layer devices, demonstrating improved response speeds with t_on_ at 22.3 ms and t_off_ at 14.2 ms compared to the contrastive display. The superior response speed of inkjet-printed dielectric layer devices can be attributed to the protective role of the dielectric layer over the hydrophobic layer. This protective effect helps prevent defects in the hydrophobic layer, leading to faster response times. In essence, the dielectric layer serves as a crucial component in enhancing the overall performance of EWDs, contributing to faster transitions between the “On” and “Off” states and, consequently, improved video display capabilities.

## 5. Conclusions

In this article, a novel EWD device containing a patterned dielectric layer based on inkjet printing was developed. A 1.25 μm thick layer of HN-1901 photoresist layer was successfully printed between the bottom electrode and the hydrophobic surface as the crucial dielectric and insulating layer. The inkjet printing parameters were explored experimentally, and the effect of the drop spacing and driving voltage on the film thickness, morphology and uniformity was demonstrated. Furthermore, the influence of the printed dielectric layer on the electrical and optical performance on device level, such as the leakage current, open voltage, and response time, were studied. The results proved that this inkjet printed HN-1901 layer could greatly hinder the dielectric failure of the EWD device by decreasing the breakdown voltage and preventing pinhole. The other benefits include a two times quicker pixel response. As for the drawback, this additional printed dielectric layer indeed led to an slightly increased driving voltage, which can be improved through the material selection and film-thickness optimization in the future work. Overall, the inkjet printing technology was proven to be effective and offered a valuable approach for fully printed electrowetting displays. The influence of the printed dielectric layer on the leakage current, open voltage and response time were studied. The results proved that the EWD device with inkjet printing dielectric layer had lower breakdown voltage and quicker pixel response.

## Figures and Tables

**Figure 1 nanomaterials-14-00347-f001:**
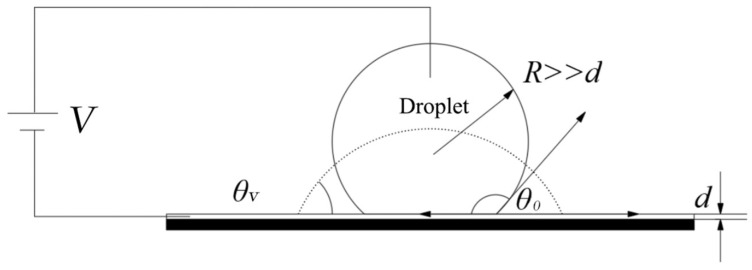
Basic schematic diagram of an EWOD device.

**Figure 2 nanomaterials-14-00347-f002:**
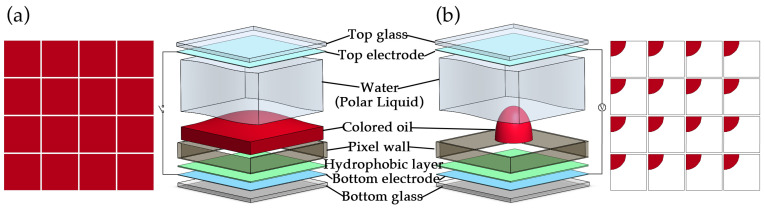
Pixel structure in an Electrowetting Display Panel. (**a**) is the state of the colored oil spread without applying voltage, and (**b**) is the state of the colored oil shrunk to the corner of a pixel wall after applying a driving voltage.

**Figure 3 nanomaterials-14-00347-f003:**
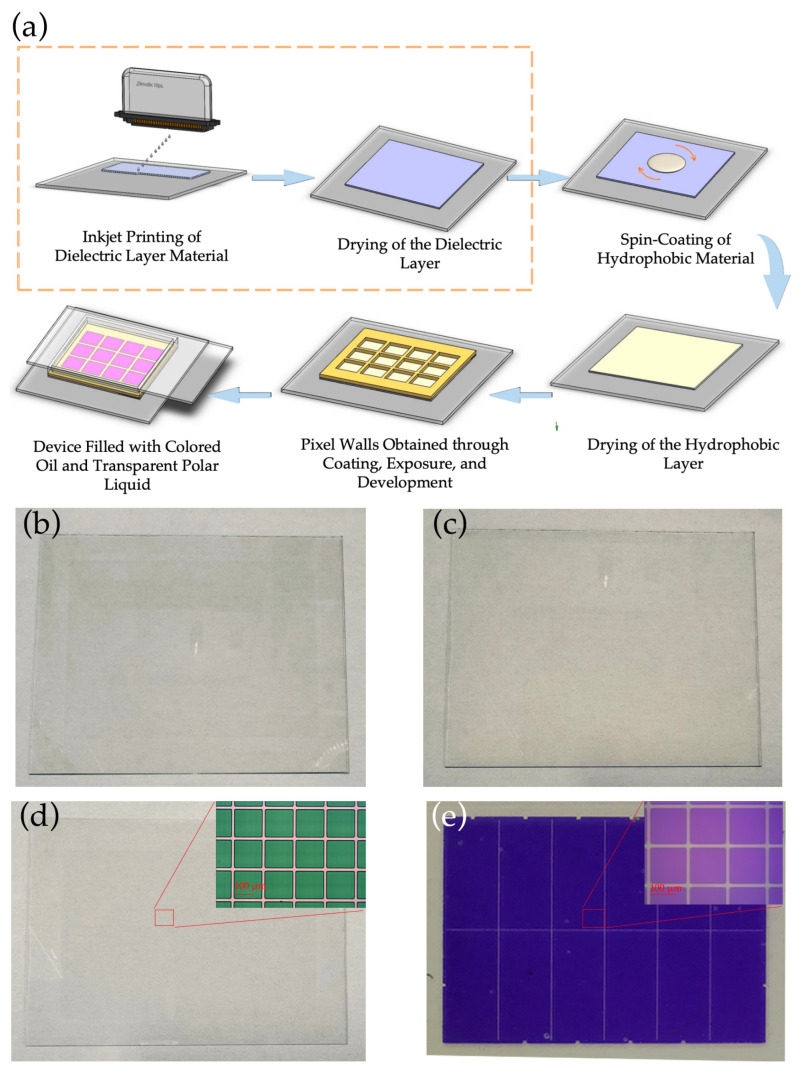
(**a**) EWD device fabrication process flowchart; (**b**) is the inkjet-printed dielectric layer; (**c**) is the hydrophobic layer obtained by spin-coating; (**d**) depicts the pixel wall after photolithography; and (**e**) illustrates the whole electrowetting device after oil and polar liquid filling.

**Figure 4 nanomaterials-14-00347-f004:**
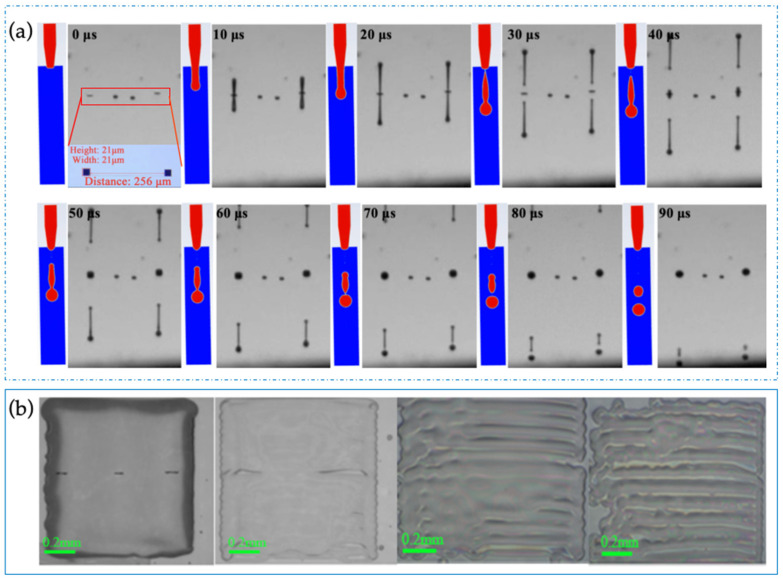
(**a**) The simulated process of droplet printing formation and the actual printing of droplets; (**b**) correlation between varied ink drop spacing and film uniformity at 20 μm, 25 μm, 30 μm, and 40 μm, respectively.

**Figure 5 nanomaterials-14-00347-f005:**
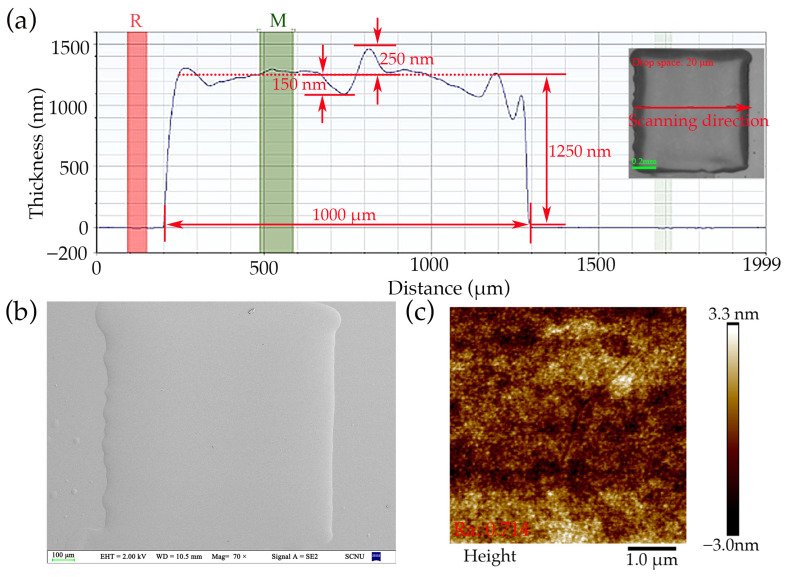
Surface morphology analysis of dielectric layer obtained by inkjet printing with 20 μm drop spacing (**a**) Analysis of film thickness data under Dektak; (**b**) an SEM image magnified 70 times; (**c**) an AFM top view image of dielectric layer; the scanning range is 5 × 5 μm, Ra is 0.714 nm.

**Figure 6 nanomaterials-14-00347-f006:**
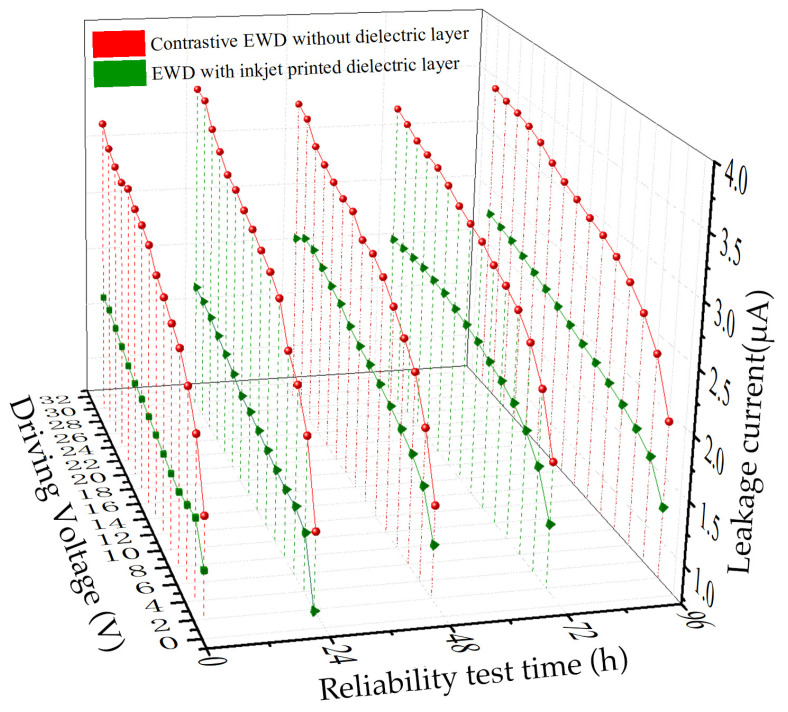
The comparison of leakage current of inkjet printed dielectric layer EWD device and contrastive EWD device after 0 h, 24 h, 48 h, 72 h, and 96 h, respectively.

**Figure 7 nanomaterials-14-00347-f007:**
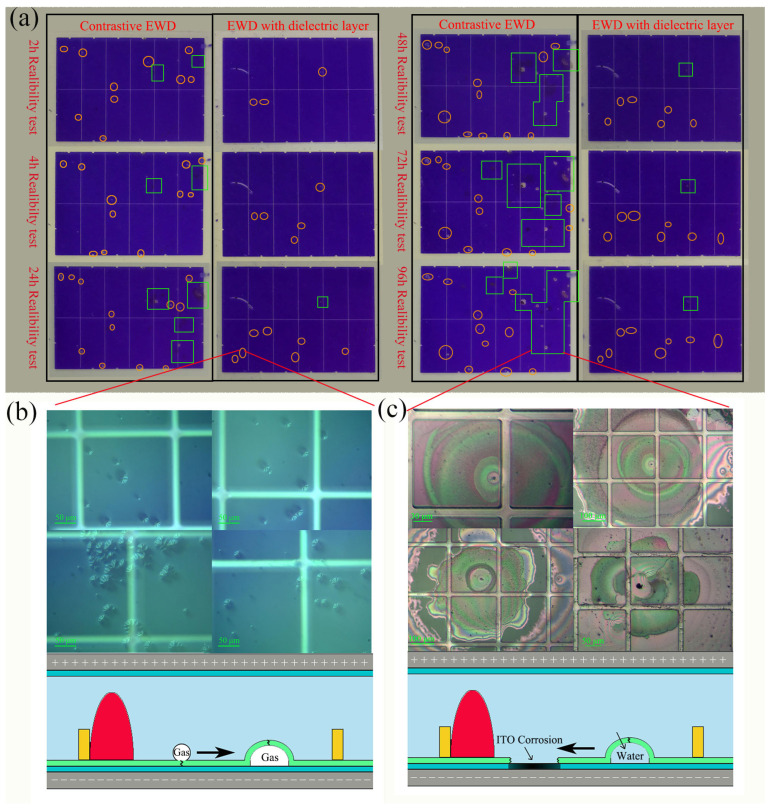
(**a**) Defects of two kind devices after 0 h, 24 h, 48 h, 72 h, and 96 h reliability test, the square represented the defect of electrode corrosion, and the circle represented the bubble defect. (**b**) The bubble defects under microscope. (**c**) The ITO corrosion defects under microscope.

**Figure 8 nanomaterials-14-00347-f008:**
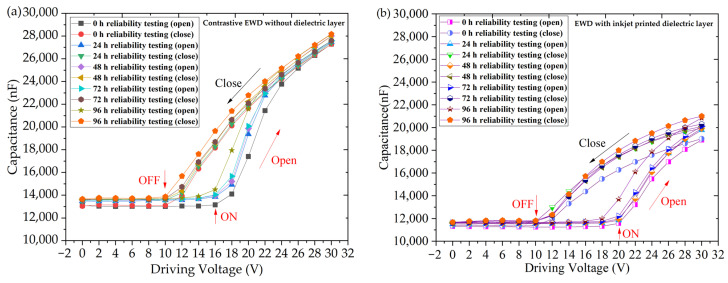
Capacitance–voltage curves of two type EWDs after 0 h, 24 h, 48 h, and 96 h reliability tests. (**a**) is the Capacitance-voltage curves of contrastive EWD without a dielectric layer, and (**b**) is the Capacitance-voltage curves of EWD with inkjet printed dielectric layer.

**Figure 9 nanomaterials-14-00347-f009:**
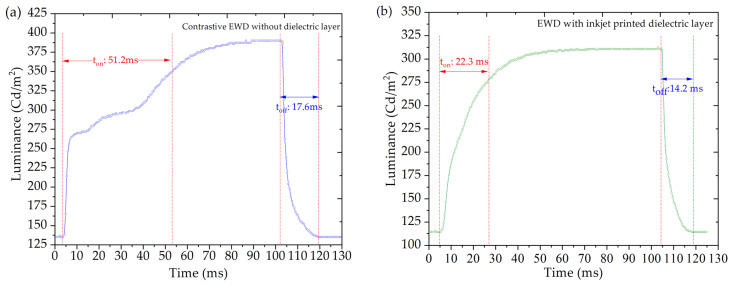
Response time curves of two devices under a 30 V driving voltage. (**a**) is the response time curve of contrastive EWD without a dielectric layer, and (**b**) is the response time curve of EWD with inkjet printed dielectric layer.

**Table 1 nanomaterials-14-00347-t001:** Physical parameters of printing materials and inkjet printing setup parameters.

Material Parameters	Value	Print Parameters	Value
Material	HN-1091	Nozzle dia. (μm)	21
Solvent	PGMEA	temperature (°C)	45
Concentration	19%	Frequency (kHz)	1.5
Density (kg/m^3^)	1010	Voltage (V)	35
Viscosity (mPa·s)	3.783	velocity (m/s)	8
Surface tension (mN/m)	11.34	Z (Oh^−1^)	4.1

## Data Availability

Data are contained within the article.

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
