# Peer review of "Inkjet-Printed Dielectric Layer for the Enhancement of Electrowetting Display Devices"

_nanomaterials, 2024, doi:10.3390/nano14040347_

Round 1

Reviewer 1 Report

Comments and Suggestions for Authors

Dear Authors,

You will find below some remarks from your excellent scientific work:

1/ Introduction part :

->  syntax error : Line 52 ( dewetting of)  and Line 72  "and spin"

-> please, the introduction must be improved at the end, because  it is difficult  to understand what is the objective of your work !! 

2/Principle Part 

-> Tittle ? Principle of what ? perhaps to choose another title ! 

-> Figure 1 : please additional  explanations are requested directly on the graph

-> equation 1 : please you must define all variables, this is not the case here!

-> equation 2 : VEW  be carreful ! same sigle in equation 4 and 5 !!! 

-> lines from 116 to127  :no references ? 

-> syntax error  : line 137 ( red color )  and line 138 FRom(m)

-> please you must clarify the objectif of this part ! not clear for me ! 

-> Please, for all equations, do not forget the unity (for each parameter)!  

3/ Device fabrication part 

-> Line 164  : Figure 2 not figure 1  ! 

-> FRom line  145 to 162 => please  could you give us more informations on  figure 2 ? perharps, you can write dielectric layer , etc...!

-> 3.4 ? EFD ? please what is EFD ? 

-> to well understand, you could modify figure 2  or insert another figure  with the process fabrication ! because it is no easy to understand

-> it is possible also to show a picture a your device ! 

4/ Results and discussion 

->  Why is there no information on the roughness of your resin layer? to clarify

-> did you measure the dielectric permittivity of your resin layer ?  why? to clarify

-> LIne 281 : syntax error "100 µ m "  

-> LIne 313 : Syntax error .. " between "display and To"

-> figure 5 => not esay to read the legend axis ( must be corrected ! )  and you must clarify the legend AF ? HN + AF ? !!

-> 

-> Figure 7 :  please improve the legend uder the graph  what is (a) ? what is (b) ?  becareful on axis legend nF would be better ! 

-> between line 384 and 393 :  missing informations ! Conventional EWD which one? lack of information on the conditions used to obtain the curves ( Fig 8) !!! 

-> Figure 8 :  please improve the legend uder the graph  what is (a) ? what is (b) ?  conventionnal and dielectric layer  !! 

5/ Conclusion 

-> a little short! it needs to be improved!  

-> don't hesitate to give important results!

-> question : do you have an idea of the humidity impact on your resin layer ? do you see any problems as layer degradation due to the voltage ? 

Author Response

Dear reviewer and editor:

Thank you very much for taking the time to read and modify my article. Thank you for your valuable suggestions. You made comprehensive corrections to the content, research methods, and results of my paper. This significantly contributes to improving the quality of my paper.

I carefully studied the reviewers' comments and carefully revised the paper according to the suggestions, as follows:

Please find the attachment for the revised PDF paper.

1/ Introduction part :

->  syntax error : Line 52 ( dewetting of)  and Line 72  "and spin"

Reply: Corrected, please find it in line 52 and line 70

-> please, the introduction must be improved at the end, because  it is difficult  to understand what is the objective of your work !! 

Reply: Modified according to your knowledge, please find line 72-78

2/Principle Part 

-> Tittle ? Principle of what ? perhaps to choose another title ! 

Reply: Corrected, please find the line 79

-> Figure 1 : please additional  explanations are requested directly on the graph

Reply: Additional explanation has been provided regarding the content of Figure 1, please find the line 84 to 91

-> equation 1 : please you must define all variables, this is not the case here!

Reply: It has been modified, please find from line 99 to 103

-> equation 2 : VEW  be carreful ! same sigle in equation 4 and 5 !!! 

Reply: modified

-> lines from 116 to127  :no references ? 

Reply: References added from line 148 to 154, please find the references from line 582-589

-> syntax error  : line 137 ( red color )  and line 138 FRom(m)

Reply: red color was corrected, please find it line 165, Fromm is a name, it was modifed to Fromm J E, for the convenience of distinguishing

-> please you must clarify the objectif of this part ! not clear for me ! 

Reply: improved, please find the line129 to 136 and line171 to 174, aims to explain why dielectric layers are added, why inkjet printers are used and the principles of printability used to guide printable material selection.

-> Please, for all equations, do not forget the unity (for each parameter)!  

Reply: I added SI units for each parameter in their first definition, please find the line 94,line 96, line 100, line 101, line 114 and line 165

3/ Device fabrication part 

-> Line 164  : Figure 2 not figure 1  ! 

Reply: corrected.

-> FRom line  145 to 162 => please  could you give us more informations on  figure 2 ? perharps, you can write dielectric layer , etc...!

Reply: added, please find the line 178 to 187

-> 3.4 ? EFD ? please what is EFD ? 

Reply: correct EFD to EWD, they are same thing, really sorry for that mistake, becase we try to verify our EWD device and Amazon EWD, we usually call our device as EFD device.

-> to well understand, you could modify figure 2  or insert another figure  with the process fabrication ! because it is no easy to understand

Reply: added as Figure 3

-> it is possible also to show a picture a your device ! 

Reply: added in Figure 3

4/ Results and discussion 

->  Why is there no information on the roughness of your resin layer? to clarify

Reply: added in Figure 5c, the Ra is 0.714nm.

-> did you measure the dielectric permittivity of your resin layer ?  why? to clarify

Reply: Sorry, I didn’t measeure the dielectric permittivity, because it is one of the material properties and has nothing to do with the thickness. The dielectric permittivity of the material used in this article is 4

-> LIne 281 : syntax error "100 µ m "  

Reply: corrected, please find the line 344

-> LIne 313 : Syntax error .. " between "display and To"

Reply:corrected, please find the line 377

-> figure 5 => not esay to read the legend axis ( must be corrected ! )  and you must clarify the legend AF ? HN + AF ? !!

-> Reply:corrected, please find figure 6, and the legend was also modified.

-> Figure 7 :  please improve the legend uder the graph  what is (a) ? what is (b) ?  becareful on axis legend nF would be better ! 

 Reply:corrected, please find figure 8.

-> between line 384 and 393 :  missing informations ! Conventional EWD which one? lack of information on the conditions used to obtain the curves ( Fig 8) !!! 

 Reply:corrected, please find the line 445 to 452

-> Figure 8 :  please improve the legend uder the graph  what is (a) ? what is (b) ?  conventionnal and dielectric layer  !! 

 Reply: Modified the legend, please find the line 452 to 456

5/ Conclusion 

-> a little short! it needs to be improved!  

-> don't hesitate to give important results!

Reply: modified according to your suggestion, please find the improved version from line 468 to line 485.

-> question : do you have an idea of the humidity impact on your resin layer ? do you see any problems as layer degradation due to the voltage ? 

Reply: Humidity does not have much impact on the resin layer, because there is water as a liquid in our device. Because the current driving voltage is relatively low, no obvious defects due to voltage have been seen for the time being.

Thank you again for your review of my article. I have benefited a lot from this revision.

Reviewer 2 Report

Comments and Suggestions for Authors

The article presents a new way to create a dielectric layer for display devices based on electrowetting (EWD). The paper is interesting and can find interest among scientists and engineers. However, before publication, some minor issues should be resolved.

1. All abbreviations should be explained with first use.

2. Please check the Figure numbering and their references.

3. Authors should provide more detail about used materials, eg. what kind of oil was used, HN-1901 contains 38% solid, what is it, what is the size of solid particles, etc.

4. Authors should use SI units and units derived from them throughout the whole manuscript.

5. What is the distance between single nozzles?

6. If the authors performed a simulation in Fluent, it should be included in the manuscript.

Author Response

Dear reviewer and editor:

Thank you very much for taking the time to read and modify my article. Thank you for your valuable suggestions. You made comprehensive corrections to the content, research methods, and results of my paper. This significantly contributes to improving the quality of my paper.

I carefully studied the reviewers' comments and carefully revised the paper according to the suggestions, as follows:

Please find the attachment for the revised PDF paper.

  1. All abbreviations should be explained with first use.

Reply: Corrected, the abbreviations were explained in line 100 to line 103, the missing definaiton was added.

  1. Please check the Figure numbering and their references.

Reply: checked, Figure 1 changed to Figure 2, find it line 207. sorry for this kind of low level mistake, and subsequent figure numbers have also been checked, and the corresponding numbers in the text have also been checked.

  1. Authors should provide more detail about used materials, eg. what kind of oil was used, HN-1901 contains 38% solid, what is it, what is the size of solid particles, etc.

Reply:  the information of oil was added from line 199 to line 202, and the information of HN 1901 was added from line 275 to 277, and I changed the solid content to concentration, there is not soild particle components in solution, so I think the concentration is more suitable.

  1. Authors should use SI units and units derived from them throughout the whole manuscript.

Reply: Checked and corrected. you can find the correction in line 222, line 230, line 232, line 280, line 299,line

  1. What is the distance between single nozzles?

Reply: the distance between the nozzles is 21μm, the information was added in the line 299.

  1. If the authors performed a simulation in Fluent, it should be included in the manuscript.

Reply: The information was added from line 290 to line 296, droplet ejection simulation is a very typical case in fluent, so I only briefly explain the solver type and parameters used in this article.

Thank you again for your review of my article. I have benefited a lot from this revision.

Round 2

Reviewer 1 Report

Comments and Suggestions for Authors

Dear Author,

Thank you about the correction done and for the improvment of your article.

Sincerley yours